# Effect of Bubbling on Ignition of PMMA Slab: Change in Thermo-Physical and Thermo-Radiative Properties

**Chloë Vincent [1,2], Claire Longuet [1], Laurent Aprin [2], Pierre Slangen [3], Guillaume Rambaud [4]**
**and Laurent Ferry [1,\*]**

1   Polymers Composites and Hybrids (PCH), IMT Mines Ales, 30100 Ales, France; chloe_vincent@live.fr (C.V.);
    claire.longuet@mines-ales.fr (C.L.)
2   Laboratory for the Science of Risks (LSR), IMT Mines Ales, 30100 Ales, France; laurent.aprin@mines-ales.fr
3   EuroMov Digital Health in Motion, University of Montpellier, IMT Mines Ales, 30100 Ales, France;
    pierre.slangen@mines-ales.fr
4   CEA, DAM, GRAMAT, BP 80200, 46500 Gramat, France; guillaume.rambaud@cea.fr
\*   Correspondence: laurent.ferry@mines-ales.fr

**Abstract:** In semi-transparent polymers, ignition is not only dependent on conductive thermal transfer into the material but also on in-depth absorption of the radiation. The aim of this work was to investigate the influence of bubbling on the thermo-physical and thermo-radiative properties of PMMA and how it may affect its ignition. PMMA plates of varying thickness were exposed to the heat flux of two radiative sources with different emission spectra. Exposure was stopped after different periods of time to study bubbling kinetics and bubble size distribution by optical microscopy. Front and back surface temperatures of samples were recorded during heat exposure. The results indicate that the bubble size distribution is closely related to the temperature gradient within the sample. Steep thermal gradients lead to small-sized bubbles underneath the exposed surface, while weak thermal gradients generate a wider size distribution with in-depth bubbling. All thermo-physical quantities $k$, $\rho$ and $C_p$ were shown to decrease with increasing bubbling degree. Likewise, it was highlighted that bubbling modifies the thermo-radiative properties of PMMA, especially in the near-infrared range. Transmittance decreases while absorbance increases with a bubbling degree. The increase in the absorption coefficient was attributed to multiple scattering by bubbles that expand the pathway of radiation into the materials. It was concluded that changes in both the thermo-physical and thermo-radiative properties with bubbling were likely to account for the delay in ignition observed when using the near-infrared heating source.

**Keywords:** bubbling; PMMA; ignition; combustion; thermo-physical properties; thermo-radiative properties

## 1. Introduction

Ignition of thermoplastic polymers has been widely investigated in the literature. Many papers about this subject were dedicated to poly(methyl methacrylate) (PMMA) because it is a non-charring polymer with a well-known and simple decomposition mechanism (i.e., depolymerization) [1–4]. From the heat conduction equation, Hopkins and Quintiere [5] proposed a model describing the time-to-ignition of non-charring absorbing polymers exposed to a radiant heat source. The model considers heat convective and radiative heat losses at the surface. Predicting ignition with this model requires knowing the thermo-physical properties of the polymer (heat capacity, thermal conductivity, and density) as well as its emissivity and ignition temperature. Dealing with PMMA, it was shown that in some cases, this classical theory failed to predict ignition, especially when the polymer is exposed to high heat fluxes [6]. In-depth radiation absorption was invoked to explain this mismatch. Delichatsios et al. [7] proposed an integral equation taking into this phenomenon to describe the surface temperature of semi-transparent samples. Starting from the same energy balance equation, Jiang et al. proposed an analytical solution for

the sample temperature using Laplace transforms [2]. More recently, very similar models were used by Gong et al. [8–10], Zhai et al. [11], Lai et al. [12] or Alinejad et al. [13] to study PMMA ignition. In all cases, the absorption coefficient κ is the key parameter monitoring in-depth absorption. Jiang et al. measured the absorption coefficient and analyzed the effect of in-depth heating with surface heat loss. Equations, initial and boundary conditions are given as follows:

Energy Balance Equations for Solid Phase

$$\frac{\partial^2 \theta}{\partial x^2} - \frac{1}{a}\frac{\partial \theta}{\partial t} = -\frac{(1-r)\dot{q}''_{ext}\kappa}{k}exp^{-\kappa x} \tag{1}$$

In this equation, $\theta = T - T_0$ is the temperature rise (K), $x$ is the in-depth distance (m), $a$ is the thermal diffusivity (m$^2$·s$^{-1}$), $k$ is the thermal conductivity (W·m$^{-1}$·K$^{-1}$), $r$ is the surface reflectivity, $\kappa$ is the absorption coefficient (m$^{-1}$) and $\dot{q}''_{ext}$ is external/incident heat flux (kW).

Initial and boundary conditions

$$\begin{cases} \theta(x,0) = 0 \\ -k\frac{\partial \theta}{\partial x}\Big|_{x=0} = -h_c(T_s - T_0) - \varepsilon\sigma_{SB}(T_s^4 - T_0^4) \\ \theta(\infty,t) = 0 \end{cases} \tag{2}$$

With $T_s$ is the surface temperature (K), $T_0$ is the environment/initial temperature (K), $h_c$ is the surface convective heat transfer coefficient (W·m$^{-2}$·K$^{-1}$), $\varepsilon$ is the emissivity and $\sigma_{SB}$ is the Stefan-Boltzmann constant ($5.670373 \times 10^{-8}$ W·m$^{-2}$·K$^{-4}$).

The analytical solutions of the model are presented in [2] considering a few assumptions as all non-reflected incident heat flux enters the solid by radiation. They showed that in-depth radiation is the primary cause of delayed ignition time for black PMMA samples. Linteris et al. [14] developed two experimental methods to determine the absorption coefficients of various polymeric materials. Some polymers found that the absorption coefficient varied with material thickness. Boulet et al. [15] have highlighted that PMMA slab exhibits a strong non-grey behavior that depends on the radiative source that is used to provoke polymer ignition [16,17]. They have carefully measured the optical and radiative properties of PMMA slabs and used them to simulate polymer heating.

Boulet et al. mentioned in their conclusion that optical properties may be affected by bubble formation [17]. In another paper, Mindykowski et al. [18] mentioned that in-depth radiation may be absorbed by bubbles at the sample surface and then inward re-emitted. Bubbling is a phenomenon affecting thermoplastic polymers when heated above pyrolysis temperature [19]. Several authors proposed a model to describe the nucleation, growth and deformation of bubbles submitted to thermal irradiation [20–22]. Polymer heating is likely to produce gasification deep in the material. Thus, it leads to the formation of internal bubbles coming up to the surface. Bubbling not only enables volatile gas transport to the flame but also modifies the local thermal conductivity as well as the internal absorption of radiation. Oztekin et al. [23] investigated the fire performance of poly (aryl ether ether ketone) (PEEK) exposed to moisture prior to the test. They observed that wet samples exhibited a shorter time to ignition. They evidenced that moisture generates an additional and early bubbling with a high number of small bubbles. It was assumed that bubbles hinder in-depth absorption by modifying optical properties, and thus, bubbled PEEK behaves like a surface-absorbing material. Safronava et al. [24] studied the fire behavior of different polymers containing moisture. They evidenced that the release of water as steam generates strong bubbling that modifies the time to ignition. They concluded that bubbling creates a foam structure with modified thermo-physical properties. Thanks to a simulation with Thermakin software (a computational tool for modeling polymer pyrolysis and combustion behaviors developed by the Federal Aviation Administration FAA), they were able to say that it can justify the observed discrepancy. The role of bubbling in modifying heat transfer was also observed by Fina et al. [25] when studying the ignition

of polypropylene/nanoclay composites. They claimed that infrared radiation may be scattered in various directions by bubbles, resulting in a lower transmitted flux. The same assumption was made by Linteris et al. [14] to justify absorption coefficients measured at room temperature may vary when increasing temperature. Recently, Hossain et al. showed that bubbling may also affect the flame spread rate by modifying the effective thermal conductivity of the condensed phase [26]. The authors determined the bubble size distribution in the molten zone after the flame spread.

Although the role of bubbling on ignition was often invoked, it has never been investigated as such. In this paper, we studied in detail the influence of the bubbling phenomenon on the ignition of PMMA slabs by testing various factors such as sample thickness, nature of the radiative source, exposure time and incident heat flux. Bubbling was characterized, and its effect on the thermo-physical and thermo-radiative properties of the polymer was investigated.

## 2. Materials and Methods

### 2.1. Radiation Sources and Fire Tests

#### 2.1.1. Radiant Panel

An experimental device, RAPACES (RAdiant PAnel Concentrator Experimental Setup), was designed at IMT Mines Alès to study the combustion of samples exposed to radiative heat fluxes up to 80 kW/m$^2$ $\pm$ 2 kW/m$^2$. This setup has already been described in a previous paper [27] and can be seen in Figure S1 in the Supplementary Material section. The radiative source consists of two 60 kW panels with a total emitting surface of 1 m$^2$ corresponding to a maximum emitted heat flux of 120 kW/m$^2$ $\pm$ 2 kW/m$^2$. Each panel is equipped with a series of 3 kW short-wave infrared (IR) halogen lamps. Each incandescent lamp consists of a tungsten filament sealed into a quartz tube that is filled with a complex halogen gas. The lamp spectrum varies in the range between 0.5 µm (visible) to 4.5 µm (far IR) with a maximum intensity of around 1.2 µm, which corresponds to a color temperature of 2400 K when panels reach their highest power [17]. The emission spectrum may vary with the set heat flux, which is monitored by the current flowing through the filament. The test consists of submitting a vertical plate with a surface area of 10 $\times$ 10 cm$^2$ (thickness in the range of 2 to 20 mm) to the heat flux generated by IR tungsten lamps. In those experiments, piloted ignition is initiated at the top of the sample (the igniter location was kept constant). Ignition was visually detected with an accuracy of $\pm$1 s.

#### 2.1.2. Epiradiator

The second source to be used was an epiradiator (Figure S2 in the Supplementary Material section). The device consists of a coil heater placed into a hemispherical, non-grey silica casing. Horizontal 5 $\times$ 5 cm$^2$ plates (thickness in the range of 2 to 20 mm) were exposed to the radiant source for various periods of time under a heat flux of 27 kW/m$^2$ $\pm$ 2 kW/m$^2$ or 31 kW/m$^2$ $\pm$ 2 kW/m$^2$. The epiradiator can be moved away from the sample thanks to a pneumatic actuator in order to quickly stop heat exposure. The emission spectrum of epiradiator is in the mid-infrared (i.e., 1.8 to 11 µm), and for the chosen heat fluxes, the maximum intensity was around 2.1 µm, which corresponds to a color temperature of 1073 K [28]. In order to study the effect of different parameters on bubbling (exposure time, sample thickness, heat flux), the bubbling onset times $t_b$ as well as the time-to-ignitions $t_{ig}$ were first determined. Ignition was visually detected with an accuracy of $\pm$1 s. Then, different samples were exposed to the same heat flux during varying periods of time, including between $t_b$ and $t_{ig}$, in order to study the bubbling kinetics. After the chosen exposure period, irradiation was quickly stopped so that bubbles were frozen into the polymer due to its rapid cooling.

#### 2.1.3. Cone Calorimeter

The PMMA reaction-to-fire characterization was carried out with the FTT (Fire Testing Technology) cone calorimeter according to ISO 5 660 Standard [29]. Tests were carried

out with a piloted ignition. The ignition spark is positioned above the sample up to ignition. The HRR is determined using the oxygen consumption technique, considering that 13.1 MJ is released by kg of oxygen consumed with an accuracy of $\pm5\%$ [30]. The sample dimensions were $10 \times 10$ cm$^2$ and 4 mm thick. Specimens are positioned in a sample holder containing a refractory fiber insulating at the backside. In this configuration, the exposed surface is 88.4 cm$^2$. The conical resistance emits in the middle infrared range. According to Boulet et al., at 50 kW/m$^2$, the cone heater is equivalent to a blackbody at 1150 K [28]. Ignition was visually detected with an accuracy of $\pm1$ s.

### 2.2. Measurements and Characterization

### 2.2.1. Temperature Measurement

The front and rear surface temperatures of PMMA samples were measured by different means. Type K thermocouples (measurement up to 1200 °C $\pm$ 0.5 °C) were positioned in contact with the sample surface. An Optis CT laser pyrometer was used for non-contact thermal measurement during the epiradiator test with a precision of $\pm1$ °C. The apparatus operates in the spectral range from 8 to 14 µm. It was located circa 15 cm from the sample surface in order to be at its focal distance. For some experiments with the radiant panel, the temperature was measured using a SC4000 HS infrared camera from FLIR Systems. The SC4000 camera has a resolution of $320 \times 256$ pixels, corresponding to 81,920 IR thermometers used simultaneously. The detection range of this camera is 3–5 µm. For all temperature measurements with IR thermal sensors, PMMA emissivity was supposed to be 0.86 (value given by the supplier) in the spectral operating range.

### 2.2.2. Thermal Diffusivity Measurement

The thermal diffusivity $a$ (m$^2$·s$^{-1}$) was measured using the Linseis XFA 600 Xenon Flash device, based on the flash method (a non-contact method). The sample is placed on a sample holder located in an oven. The furnace is maintained at a predetermined temperature. At this temperature, the lower part of the sample is irradiated with a programmed energy pulse (xenon-flash lamp). This energy pulse results in a temperature rise in the sample. The resulting temperature rise of the rear surface is measured by a very sensitive high-speed IR detector. Thermal diffusivity is computed from temperature rise versus time curve. Thereafter, the thermal conductivity of the material can be determined using the following equation:

$$a = \frac{k}{\rho \times C_p} \tag{3}$$

With $k$ is the thermal conductivity (W·m$^{-1}$·K$^{-1}$), $\rho$ is the material density (kg·m$^{-3}$) and $C_p$ is the specific heat (J·K$^{-1}$·kg$^{-1}$).

### 2.2.3. Radiative Properties Measurement

Hemispherical spectral reflectance $R(\lambda)$ and transmittance $\tau(\lambda)$ were measured using a Jasco V-670 spectrophotometer in the range 190 nm to 2500 nm and using a Tensor 27 Bruker spectrometer in the range 1.3 µm to 14 µm, both equipped with integrating spheres.

The apparent absorptance $\alpha$ of a sample irradiated by a source characterized by a spectral energy distribution E($\lambda$) is calculated from spectral transmission $\tau(\lambda)$ and spectral reflection $R(\lambda)$ according to Equations (4)–(6). The spectrum is discretized by the same spectral step $\Delta\lambda$ (circa 300 nm) between the initial wavelength of the spectrum $\lambda_i$ and the final wavelength $\lambda_f$.

$$\alpha = 1 - \tau - R \tag{4}$$

$$\tau = \frac{\sum_{\lambda_i}^{\lambda_f} E(\lambda)\cdot\Delta\lambda\cdot\tau(\lambda)}{\sum_{\lambda_i}^{\lambda_f} E(\lambda)\cdot\Delta\lambda} \tag{5}$$

$$R = \frac{\sum_{\lambda_i}^{\lambda_f} E(\lambda) \cdot \Delta\lambda \cdot R(\lambda)}{\sum_{\lambda_i}^{\lambda_f} E(\lambda) \cdot \Delta\lambda} \tag{6}$$

In our study, the spectral energy distribution $E(\lambda)$ corresponds to the distribution of the RAPACES spectrum, i.e., that of a black body at 2400 K.

### 2.2.4. Imaging

A Zeiss stereo microscope was used for the observation of samples containing bubbles. Magnifications ranging from 6 and 18 were utilized. Samples (S = 35 mm$^2$) were illuminated by low-angled light for better contrast of bubbles.

The technique of shadowgraphy was also used to evidence, count and size bubbles. The principle of shadowgraphy consists of using the spatial variations of the optical index n(T) as a spatial dependence tracer of the temperature field T [31], but in the present paper, the use of shadowgraphy is not to highlight a temperature field. This technique requires a high acquisition frequency camera and a backlight illumination system that allows, in our case, visualization and quantification of bubbles in the PMMA sheet. The analysis and image processing software Aphelion$^{TM}$ version 4.0 was used to quantify the bubbles. The principle consists of ten steps:

- Image masking on the desired region of interest
- Color plane extraction to HSV plane
- Smoothing Gaussian by 5 × 5 kernel
- Highlight details by 3 × 3 convolution kernel (central weight 10, external weight −1)
- Threshold on dark or bright objects depending on the illumination setup: manual threshold or Clustering threshold depending on the bubble spacing (manual if touching bubbles)
- Proper Open 3 × 3 kernel to separate touching bubbles
- Filling holes to get the full area of the bubbles
- Removing borders objects
- Filtering objects: (i) user selected size, or (ii) circularity with Heywood factor criterion, to remove touching bubbles if the user selected
- Classification

### 2.3. Material

Experimental measurements were conducted with PMMA sheets. According to Bal and Rein [6], the best material for investigating the ignition of a solid is PMMA; many experimental data, numerical studies, and properties are listed in the literature. PMMA used in this study is a clear PMMA (AbaquePlast, Stains, France) slab, and the thermophysical properties were determined experimentally in the laboratory (at 25 °C) using a 4 mm thick sample. These values are listed in Table 1, and values found in the literature are compared [32–36].

**Table 1.** Set of experimental and numerical parameter values for clear PMMA slab at 25 °C.

| Parameters | Experimental Values * (Thickness = 4 mm) | Theoretical Values |
|---|---|---|
| Density $\rho$ (kg·m$^{-3}$) | 1190 | 1170–1200 |
| Thermal conductivity $k$ (W·m$^{-1}$·K$^{-1}$) | 0.20 | 0.167–0.251 |
| Specific heat $C_p$ (J·kg$^{-1}$·K$^{-1}$) | 1400 | 1400–1520 |
| Emissivity $\varepsilon$ | - | 0.86 |
| Thermal diffusivity $a$ (m$^2$·s$^{-1}$) | $1.2 \times 10^{-7}$ | $1.1–1.4 \times 10^{-7}$ |
| Ignition temperature $T_{ig}$ (°C) | 320 | $317 \pm 10$ |

* Experimental values are the average of at least three replicates.

## 3. Results

### 3.1. Time-to-Ignition and In-Depth Absorption

The ignition times of black PMMA (the surface of the PMMA slab is coated with graphite) and clear PMMA were measured under various irradiances and using two radiative sources (radiant panel and cone calorimeter). Then, these values were compared with the literature [5]. In Figure 1, results show that the ignition time of black PMMA is close to that of clear PMMA, up to 50 kW/m² when using a cone calorimeter as a heating source. The ignition time of clear PMMA, exposed to the radiant panel heat source, was much longer than that found with the cone calorimeter at the same irradiance. Under a given heat flux, it was also shown that the time-to-ignition of clear PMMA was dependent on the sample thickness up to 20 mm. These results indicate that PMMA absorptivity should be relatively low in the spectral range of emission of the radiant panel heating source.

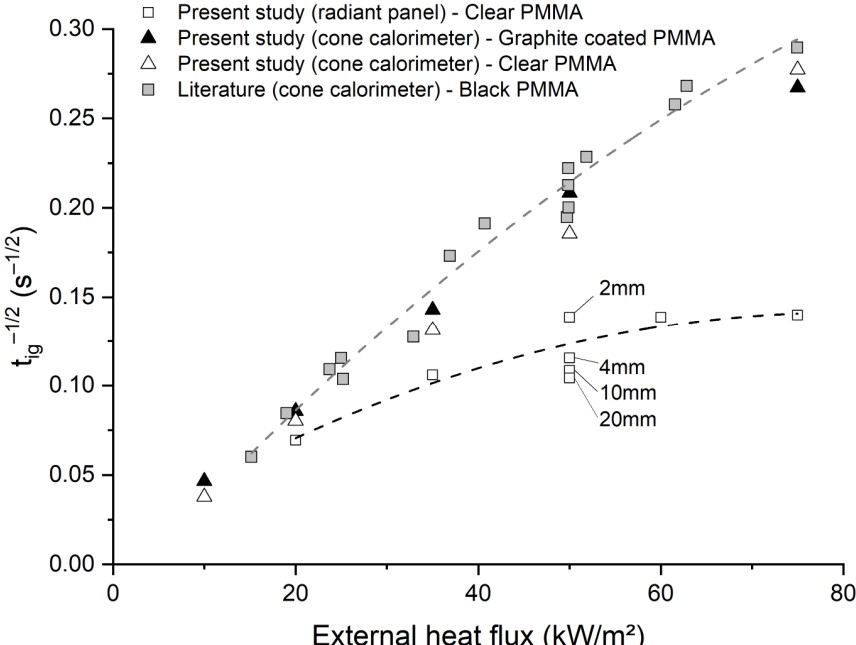

**Figure 1.** Values of versus heat flux (Experience and literature [5]—Comparison between two radiative heat sources (cone calorimeter and RAPACES).

Considering the in-depth absorption model (Equation (1)), a reverse approach was used to calculate the expected absorption coefficient $\kappa_{calc}$ of PMMA for the two sources, assuming that, at time-to-ignition, the modeled surface temperature matches with the experimental ignition temperature (i.e., 317 °C [32]). Considering that convection was relatively weak in the two fire tests (cone calorimeter and RAPACES), the convective transfer coefficient $h_c$ was taken as 10 W·m$^{-2}$·K$^{-1}$, as found in the literature [5]. The values of $\kappa_{calc}$ are given in Table 2. It was found that the absorption coefficient should be around 2050 m$^{-1}$ to obtain the correct ignition times for PMMA using the cone calorimeter. This value is in the same range as that found by Linteris et al. [14,37]. For the radiant panel, the value of the absorption coefficient should be between 280 and 2010 m$^{-1}$ to correctly simulate the time-to-ignitions.

**Table 2.** Time-to-ignition using radiant panel and cone calorimeter, calculated and experimental absorption coefficients.

| Source | Sample Thickness (mm) | Heat Flux (kW/m²) | $t_{ig}$ (s) | $\kappa_{calc}$ (m⁻¹) | $\kappa_{exp}$ (m⁻¹) |
|---|---|---|---|---|---|
| Radiant panel | 2 | 50 | 52 | 576 | 142 |
| | 4 | 50 | 75 | 360 | 35 |
| | 10 | 50 | 85 | 310 | 19 |
| | 20 | 50 | 92 | 283 | 10 |
| | 4 | 20 | 208 | 2010 | |
| | 4 | 35 | 89 | 662 | |
| | 4 | 60 | 52 | 398 | |
| | 4 | 75 | 51 | 280 | |
| Cone calorimeter | 4 | 10 | 460 | >5000 | |
| | 4 | 20 | 125 | >5000 | |
| | 4 | 35 | 51 | 2060 | |
| | 4 | 50 | 25 | 2040 | |
| | 4 | 75 | 17 | 1140 | |

In order to determine the actual absorption coefficient of PMMA in the emitting range of the radiant panel, the radiation properties of PMMA slabs were measured in the wavelength range from 190 nm to 14 µm at room temperature using the protocol described in Section 2.2.3. Figure 2 highlights that PMMA is a non-grey polymer. Moreover, it shows that the transmittance decreases with increasing thickness while the reflectance is almost constant, around 8% (Figure 2). Absorptance was deduced from transmittance and reflectance and was shown to increase with sample thickness. Since PMMA is a non-grey material, an apparent absorptance $\alpha$ was determined for each studied thickness considering the spectral absorptance and the emission spectrum of the source as described in Equations (4)–(6). An absorption coefficient $\kappa_{exp}$ was then determined in each interval as the slope of the curve ($\kappa = \frac{d\alpha}{dx}$). Table 2 highlights that $\kappa_{exp}$ varies from 10 m⁻¹ to 142 m⁻¹. These values are close to those found by Boulet et al. [15] in the NIR and visible. In any case, the value of the experimental absorption coefficient $\kappa_{exp}$ is much lower than $\kappa_{calc}$ the absorption coefficient required to correctly simulate time to ignition using the in-depth-absorption model.

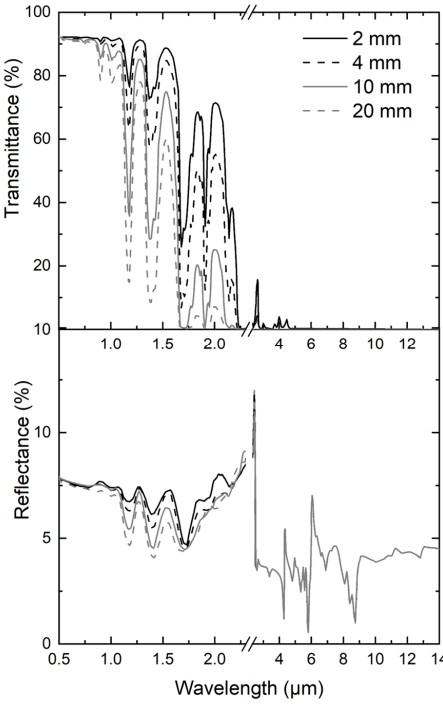

**Figure 2.** Thickness dependence of clear PMMA spectral reflectance and transmittance.

Thus, it appears necessary to invoke other phenomena that could be responsible for an additional absorption of the radiation. Transmittance measurements were performed prior to the material being exposed to the heat flux. However, when exposed to the heating source, the PMMA samples undergo an increase in temperature, inducing changes in their physicochemical and also optical properties. Since an intense bubbling of the sample was observed in the pre-ignition period, this phenomenon has been more accurately studied to understand its impact on the ignition time and the relation with the modification of the thermo-physical and/or thermo-radiative properties.

### 3.2. Study of PMMA Bubbling

The aim of this part is to study the effect of (i) exposure time, (ii) sample thickness, (iii) heat flux intensity, and (iv) the nature of the radiative heat source on the kinetics and features of bubbling.

#### 3.2.1. Effect of Exposure Time

To explore the effect of exposure time, 2 mm thick samples of PMMA were exposed to a 27 kW/m$^2$ $\pm$ 2 kW/m$^2$ radiative heat flux emitted by the epiradiator. Figure 3 shows the evolution of the front face temperature of samples according to the time of exposure to the radiative source. The curves overlap until radiation is stopped, which indicates the experiment's reproducibility. In all cases, bubbling starts after a period $t_b$ of 55 s corresponding to a bubbling temperature $T_b$ of 287 °C. Then, the temperature reaches a plateau around 320 °C, and ignition occurs after circa 90 s of exposure. This temperature is similar to the ignition temperature generally found in the literature [32].

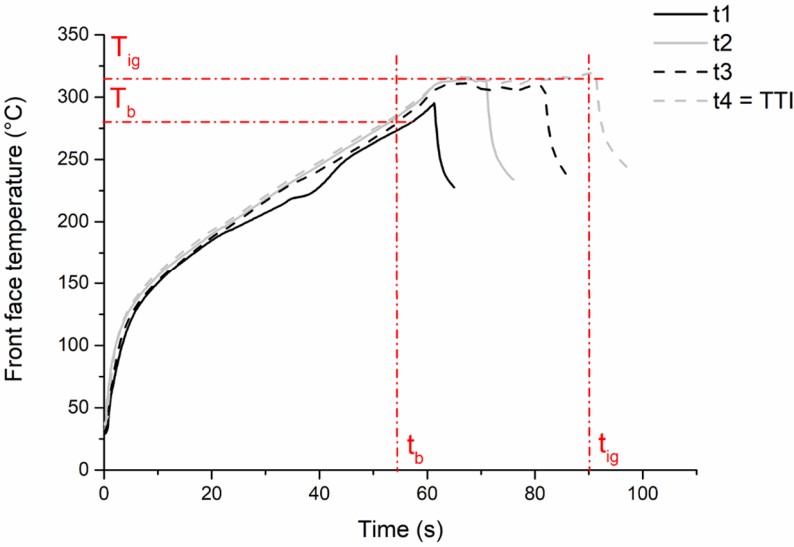

**Figure 3.** Front face temperature evolution of PMMA sample (2 mm thickness) exposed to 27 kW/m$^2$ $\pm$ 2 kW/m$^2$ during four different times of exposure.

After cooling the morphology of bubbled samples can be observed by the technique of shadowgraphy using a binocular loupe and camera as described in Section 2.2.4. Figure 4 presents the bubbling evolution according to the exposure time.

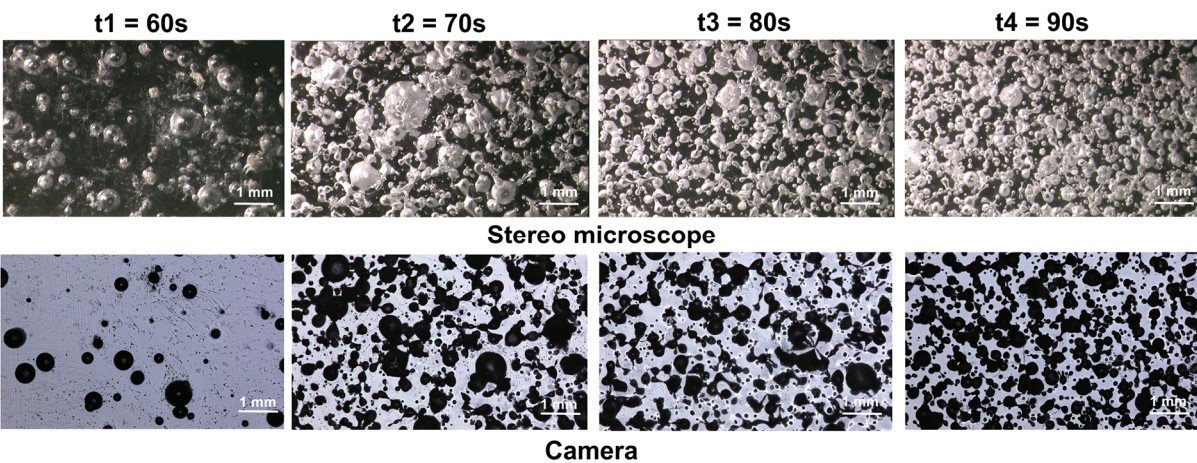

**Figure 4.** Pictures of the bubbling evolution depending on exposure time.

After image analysis, the distribution of the bubbles number, as well as the cumulative number of bubbles, were plotted according to their diameter for each exposure time tested. Measurements have been performed on a surface sample equal to 35 mm². Figure 5 shows that at the onset of bubbling, the diameter of the bubbles is relatively heterogeneous. As bubbling progresses, a bimodal distribution seems to emerge. A population of small-size bubbles (circa 50 μm diameter) is observed together with a population of larger-size bubbles (circa 200 μm diameter). At a longer exposure time close to ignition, the bimodal distribution is still observed, but the peaks are shifted to higher diameters.

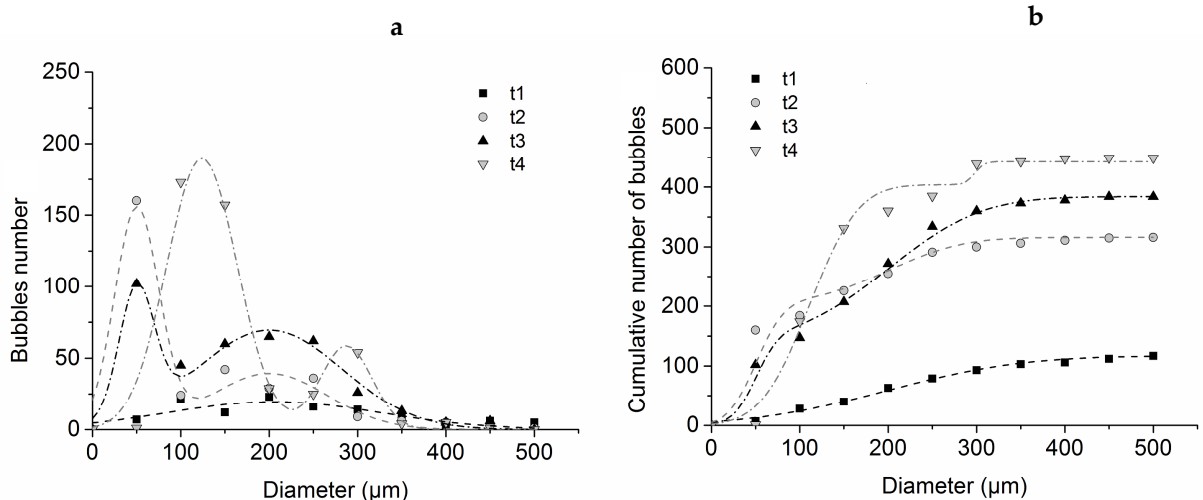

**Figure 5.** Differential (**a**) and cumulative (**b**) bubble size distribution in a 2mm thick PMMA sample exposed to 27 kW/m² ± 2 kW/m² during four different times of exposure (t1: 60 s; t2: 70 s; t3: 80 s and t4: 90 s).

Concerning the number of bubbles, it increases with increasing exposure time. Assuming that the bubbles are spherical, the total volume of bubbles can be calculated by multiplying their number by their volume. It can be deduced that the total volume increases with exposure time. This is related to the increase of the pyrolysis mass loss rate, itself related to the increase of the thermally affected layer. This can be evidenced by the temperature of the rear face of the sample, as shown in Table 3.

**Table 3.** Comparison of $t_b$, $t_{ig}$, the temperature at the front and rear surface at $t_{ig}$ and the temperature of the front surface at $t_b$ for the different tests presented in this paper.

| Radiative Source | Epiradiator | | | Radiant Panel |
|---|---|---|---|---|
| Thickness (mm) | 2 | 10 | 10 | 2 |
| Heat Flux (kW/m$^2$) | $27 \pm 2$ | $27 \pm 2$ | $31 \pm 2$ | $27 \pm 2$ |
| Onset time of bubbling $t_b$ (s) | $55 \pm 2$ | $220 \pm 2$ | $65 \pm 2$ | $100 \pm 2$ |
| Ignition time $t_{ig}$ (s) | $90 \pm 2$ | $250 \pm 2$ | $86 \pm 2$ | $120 \pm 2$ |
| Temperature of front face at $t_{ig}$ (°C) | $320 \pm 5$ | $330 \pm 5$ | $335 \pm 5$ | $218 \pm 5$ |
| Temperature of rear face at $t_{ig}$ (°C) | $269 \pm 5$ | $45 \pm 5$ | $46 \pm 5$ | $198 \pm 5$ |
| Temperature of front face at $t_b$ (°C) | $287 \pm 5$ | $325 \pm 5$ | $314 \pm 5$ | $202 \pm 5$ |

### 3.2.2. Effect of Sample Thickness

In order to study the influence of sample thickness on bubbling, 2 mm and 10 mm thick PMMA sheets were exposed to a 27 kW/m$^2$ $\pm$ 2 kW/m$^2$ heat flux at the epiradiator.

Table 3 presents the bubbling time $t_b$ and the ignition time $t_{ig}$ for each thickness. It is observed that the ignition time, as well as the onset time of bubbling, increases with increasing sample thickness. A 2 mm thick sample can be considered a thermally thin sample in which heat cannot be evacuated by the rear face and accumulates rapidly. Therefore, the temperature of the front face increases much more rapidly than in the 10 mm thick sample, and so does the temperature of the rear face, as proved by data in Table 3. It is noteworthy that the ignition temperature is constant, regardless of the thickness, whereas the bubbling temperature is slightly higher for the thick sample.

Then, the bubbling morphology has been studied, and the analysis is presented in Figure 6. It can be noted that the 10 mm thick sample exhibits a roughly unimodal distribution of bubbles. Moreover, the size of the bubbles is smaller than for the 2 mm thick sample with a mean diameter of around 70 μm. This may be due to the fact that in the 10 mm thick sample, the temperature gradient is higher than in the 2 mm thick sample, as shown in Table 3. Therefore, the bubbles are formed within a thinner layer at an elevated temperature. From the classical nucleation theory [38], the critical radius $r_c$ (radius above which nucleus becomes a stable bubble) of a bubble emerging in a superheated liquid depends on the difference $\Delta P$ between the pressure inside the bubble and the pressure in the liquid and on the surface tension σ of the liquid according to Equation (7):

$$r_c = \frac{2\sigma}{\Delta P} \tag{7}$$

In the case of thick samples, bubbles are formed in a region where σ is low and $\Delta P$ is high. Thus, their critical radius should be small. On the contrary, thin samples exhibit weak thermal gradients within their thickness. Bubbles are likely to be formed at lower temperatures with a larger critical radius.

Figure 6 indicates that the number of bubbles is doubled when the sample thickness increases from 2 to 10 mm (450 against 960 bubbles for the 35 mm$^2$ surface of counting). However, at the same time, the median diameter decreases from 117 to 67 μm, meaning that the mean bubble volume decreases by approximately a factor of 5. Therefore, it can be estimated that the total trapped pyrolysis gases decrease by more than 2.5. Hence, it appears that the pyrolysis mass loss rate decreases with increasing sample thickness, which is consistent with what was observed in the literature [39].

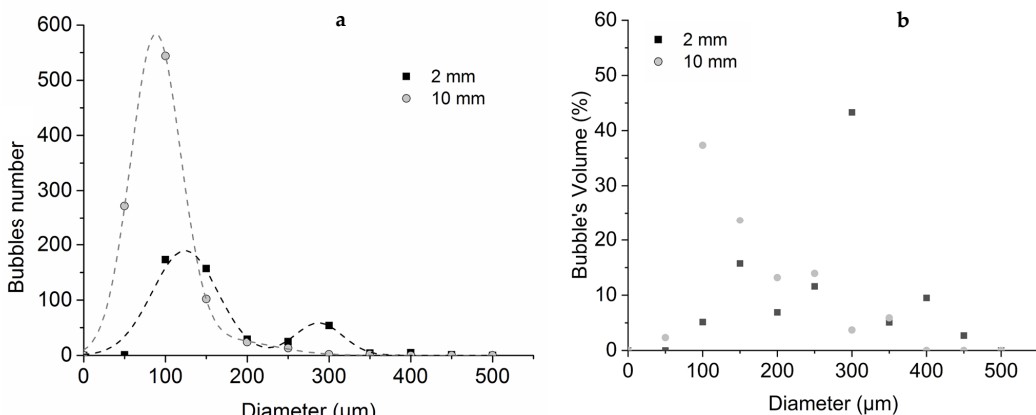

**Figure 6.** Differential bubble size (**a**) and volume (**b**) distribution in 2 mm and 10mm thick PMMA samples exposed to $27 \text{ kW/m}^2 \pm 2 \text{ kW/m}^2$ for a time near to the ignition.

Figure 7 shows that for a 2 mm thick sample, spherical bubbles are formed throughout the depth. As seen in Figure 6, two populations can be observed. The smallest bubbles seem to be located mainly near the exposed surface, while the largest bubbles are in depth. On the contrary, for the 10 mm thick sample, bubbles are essentially located near the upper surface. In this layer, small-size bubbles can be observed consistently with the distribution of Figure 6. In-depth, a small number of big bubbles are visible. This observation is consistent with the Laplace law; indeed, with increasing temperature, there is a decrease of surface tension and an increase of the pressure into the bubble. Hence, the bubble diameter is consistent with being small near the surface. Moreover, it can be noticed that some of the bubbles exhibit an elongated shape, with elongation being in the direction of the thermal gradient. This is indeed what Pickering [40] also noticed. This elongation is consistent with the Stokes law, in which the drag force decreases when viscosity decreases [22].

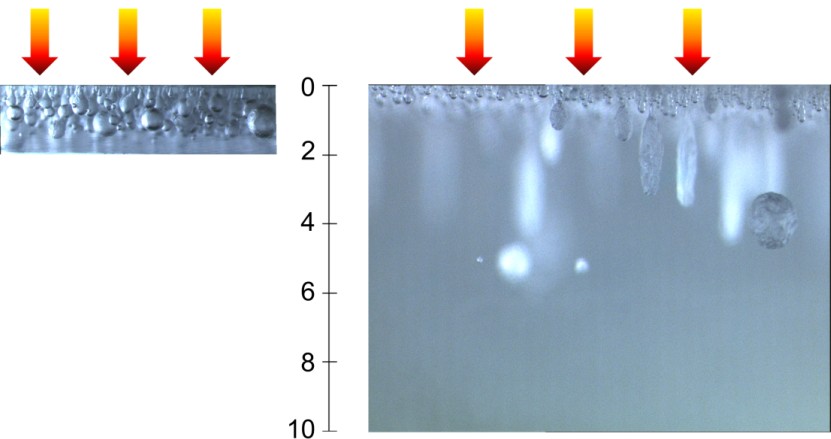

**Figure 7.** Pictures of in-depth bubbles for samples of thickness e = 2 mm (**left**) and e = 10 mm (**right**). Arrows represent the incident heat flux.

Afterward, measurements of transmittance and reflectance were performed on bubbled samples. The results show that, in two cases (2 and 10 mm thickness), transmittance decreases by 20%. Reflectance is doubled for a 2 mm thick sample (8% for t0 to 16% for t4), whereas for a 10 mm thick sample, reflectance is constant (equal to 8%).

### 3.2.3. Effect of the Incident Heat Flux

To study the influence of the heat flux on bubbling, two 10 mm thick samples have been exposed respectively to $27 \text{ kW m}^2$ and $31 \text{ kW m}^2 \pm 2 \text{ kW/m}^2$ with the epiradiator.

As expected, it was observed that the highest heat flux leads to highest heating rate of the exposed surface. Consequently, bubbling and ignition occur much more rapidly than with the 27 kW/m² heat flux as noted in Table 3.

Regarding the bubble's morphology (Figure 8), for a time close to ignition, the bubble diameter is smaller for the highest heat flux. A narrow bubble size distribution was observed, with a great part of the bubbles having a diameter of around 50 μm. Moreover, it was noticed that most of the bubbles were located within a thin layer under the surface. Our results are consistent with those of Cordova et al. [41], who noted that the bubbles are observed deeper into the sample at low external radiant flux, while at higher external radiant flux, fewer bubbles are formed in a relatively thin layer. Furthermore, these results are in agreement with those of Daikoku et al. [42], who exposed PMMA samples to different heat fluxes, ranging from 30 kW/m² to 150 kW/m². They showed that the depth of the bubbles depends on the incident heat flux: the higher the heat flux, the thinner the layer containing bubbles. They attributed this effect to the temperature gradient within the sample. High heat flux induces a steep thermal gradient; therefore, the layer in which the temperature is sufficiently high to provoke a significant pyrolysis rate is narrow and close to the exposed surface. We obtained similar results, and our measurements of the front and rear temperatures of the samples during the burning test corroborate the assumption that the thermal gradient increases with increasing heat flux.

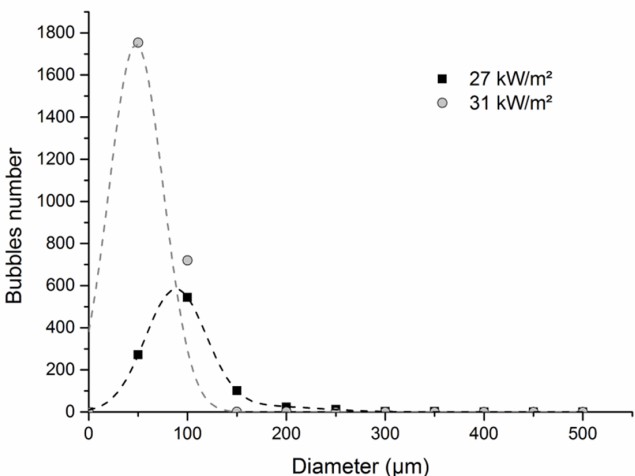

**Figure 8.** Differential bubble size distribution for 10 mm thick PMMA sample exposed to 27 kW/m² ± and 31 kW/m² ± 2 kW/m² at the epiradiator.

### 3.2.4. Effect of the Heat Source

Two heat radiative sources (electrical resistance and tungsten lamps) have been used in this study. Figure 9 shows that these sources exhibit different emission spectra. Tungsten lamps emit essentially in the near-infrared range with a maximum intensity of around 1.5 μm, while electrical resistance emits in the middle infrared range with a maximum intensity of 4.5 μm. Boulet et al. [17] studied relatively similar emitters, i.e., FPA lamp and cone calorimeter. For FPA, they also observed that the tungsten lamp emitted mainly in the near-infrared range, but they concluded that the emission cannot correctly be approximated by a simple blackbody curve. It must be weighted by the lamp's emissivity. Regarding the heating coil, they concluded that it behaves as a near-perfect blackbody with a temperature ranging from 700 to 1200 K depending on the chosen irradiance [28].

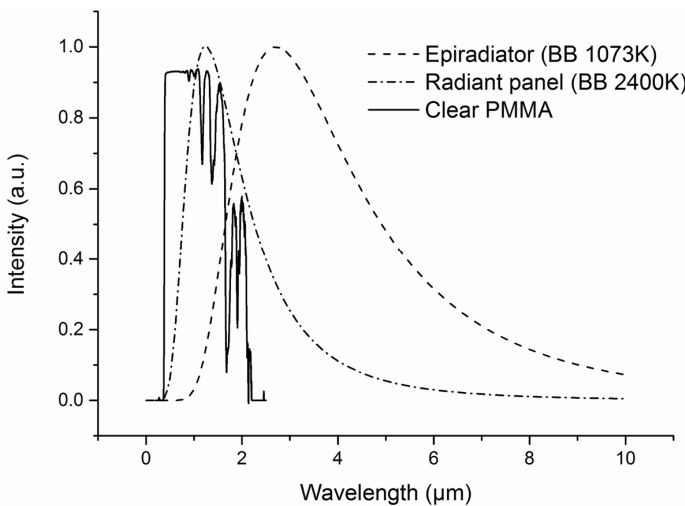

**Figure 9.** Emission spectra of epiradiator and Radiant panel and transmission spectrum of clear PMMA.

The absorption spectrum of a 4 mm thick clear PMMA sample is shown in Figure 9. PMMA is relatively transparent in the near-infrared region, whereas it absorbs more strongly radiations from 2 μm (i.e., 0% transmission for a 2 mm thick sample). Försth et al. [43] confirm that most of the energy absorbed by clear PMMA samples is around wavelengths greater than 2 μm. Bal et al. [16] showed that for a clear PMMA at a depth of 2 mm, 90% of the energy of the mid-IR source has been absorbed.

The semi-transparent character of PMMA in the emission domain of tungsten lamps will give rise to some difference in the thermal behavior, in particular as regards the bubbling phenomenon. Table 3 shows the front face temperature measurements for a 2 mm thick PMMA sample exposed to a heat flux of 27 kW/m$^2$ ± 2 kW/m$^2$ using a radiant panel (tungsten lamps) and epiradiator (electrical resistance) as heating source. The results indicate that the onset of bubbling occurs earlier when using the epiradiator, 55 s instead of 100 s with the radiant panel. Since PMMA is semi-transparent in the near-infrared range, only a small part of the incident energy emitted by the tungsten lamps is absorbed into the polymer, and thus, temperature increases more slowly. Boulet et al. [17] conclude that a transparent PMMA sample (3 cm thick) absorbs 91% of the flux received by an electrical resistance but only 32% with a tungsten lamp.

Figures 10 and 11 show the impact of the two radiative heat sources on bubbling. The number of bubbles is five times greater when the sample is exposed to an electrical resistance. The bubble diameter distribution exhibits a 125 μm median value with the epiradiator, whereas it is over 300 μm for the sample exposed to the radiant panel.

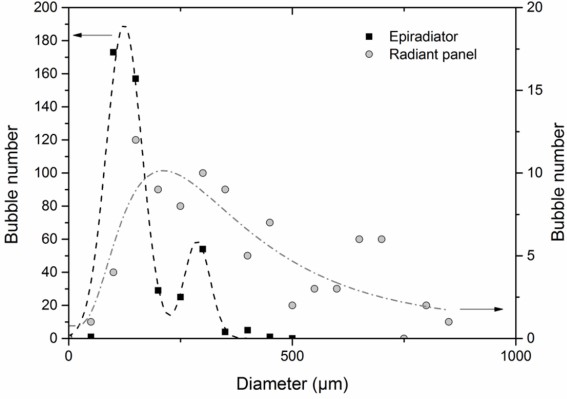

**Figure 10.** Differential bubble size distribution for 2 mm thick PMMA exposed to 27 kW/m$^2$ ± 2 kW/m$^2$ with epiradiator (right scale) and radiant panel (left scale).

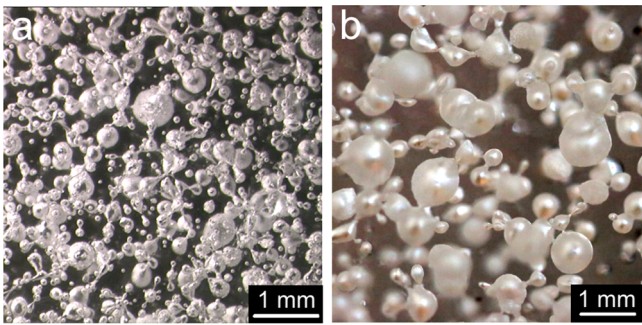

**Figure 11.** Influence of heat source on bubbling in PMMA: (**a**) epiradiator (**b**) radiant panel. Heat flux = 27 kW/m², sample thickness = 2mm.

Once again, these results can be explained by the temperature gradient within the sample (see Table 3). When bubbling starts, this gradient is much steeper in the case of epiradiator. This is in relation to the PMMA absorption coefficient with respect to the emission spectrum of the source. A greater part of the incident heat flux coming from the epiradiator is absorbed, as previously explained. Moreover, it can be noticed that bubbling starts at a lower temperature when using the radiant panel as a heating source. At this temperature, the surface tension should be relatively high, and therefore, the critical diameter for bubble nucleation is also higher.

*3.3. Evolution of Thermo-Physical and Thermo-Radiative Properties with Bubbling*

3.3.1. Thermo-Physical Properties

In order to assess the impact of bubbling on the thermo-physical properties of PMMA, 2 mm thick samples previously exposed to epiradiator at a 27 kW/m² heat flux were used. Bubbled samples with four different exposure times were assessed so that bubbling intensity was significantly varied as described in Section 3.2.1.

The bulk density $\rho$ was simply determined by accurately measuring the size and weight of bubbled samples. Figure 12 shows that as bubbling occurs, the bulk density decreases from 1.17 to 1.01 g·cm$^{-3}$. The drop in bulk density is steep, and then a plateau seems to be reached. Considering that energy is an additive quantity and that the density of gases within the bubbles is close to zero, it can be assumed that the heat capacity Cp remains constant with bubbling since it is a mass property. The thermal diffusivity was measured using a Xenon flash apparatus. The results of Figure 12 indicate that the thermal diffusivity remains almost constant circa $1.3 \times 10^{-7}$ m²·s$^{-1}$ $\pm$ 0.1 m²·s$^{-1}$, whatever the bubbling degree in the polymer. This constant value may be explained by the fact that the thermal conductivity must decrease with bubbling, thus counterbalancing the decrease in bulk density.

It must be remembered that in the model proposed by Hopkins and Quintiere [5] to predict ignition in the case of absorbing materials, the time-to-ignition depends linearly on $k \cdot \rho \cdot C_p$. Therefore, a decrease of $k$ and $\rho$ with bubbling should lead to a decrease in the time-to-ignition, whatever the source. This could be one reason for the mismatch in time-to-ignition observed with clear PMMA exposed to the radiant panel. Nevertheless, early ignition should also be observed with a mid-infrared source (cone calorimeter or epiradiator), but this was not the case. It must be added that these thermal physical quantities must be considered with caution (i) because they have been determined at room temperature and they surely vary when temperature increases and (ii) because bubbling does not start with irradiation but only after a given period of time.

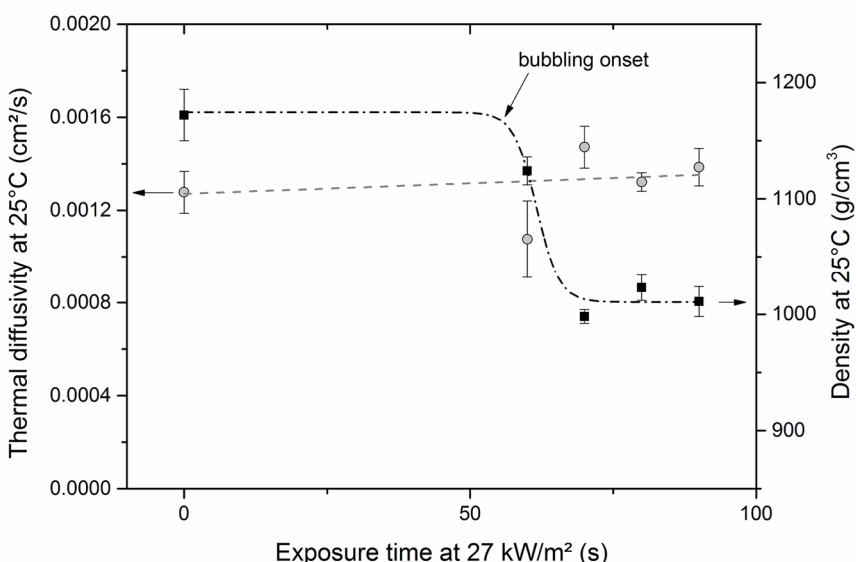

**Figure 12.** Bulk density (square, right scale) and thermal diffusivity (circle, left scale) of 2 mm thick bubbled PMMA samples.

### 3.3.2. Thermo-Radiative Properties

To check if bubbling may have an impact on the thermo-radiative properties in the range where PMMA is semi-transparent (i.e., in the near IR), 2 mm thick virgin and already bubbled PMMA samples were used. These samples were exposed once again to a $50\,\text{kW/m}^2$ heat flux produced by the radiant panel. The transmitted heat flux was measured as soon as the halogen lamps were switched on using a total fluxmeter. The value obtained $I$ was divided by the intensity of the heat flux without any sample $I_0$. Figure 13 shows that the transmitted heat flux is attenuated when the samples contain bubbles. The attenuation just before ignition seems to be approximately the same regardless of the conditions that have been used to generate the bubbling.

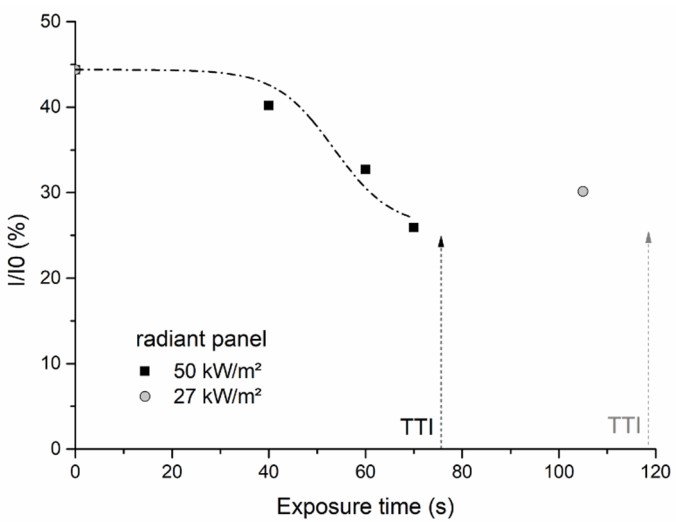

**Figure 13.** Attenuation of the heat flux transmitted by bubbled PMMA samples (TTI = time-to-ignition).

In order to obtain a more accurate assessment of the change in thermo-radiative properties with bubbling, reflectance, transmittance and absorptance of various bubbled samples were measured as depicted in Section 2.2.3. It was shown that the reflectance is around 8% and tends to slightly increase with bubbling (results not reported here).

Figure 14 highlights that the transmittance decreases from 90% at the initial time to 65% near the ignition time while, at the same time, the absorptance increases by 20%. The main difference lies in the moment when the change of properties occurs. This moment is related to the appearance of bubbling itself and the heating rate on the sample. It can be noticed that regardless of the source and the thickness of the sample, the final values of transmittance and absorptance are almost the same. Hence, bubble size distribution does not seem to play a major role in the final results. It may be assumed that the small bubbles located underneath the exposed surface and found in all samples could be responsible for the scattering of NIR radiations, as has been shown in other weakly absorbing materials [44]. According to the Mie theory, bubbles in the radius range 50–200 µm can scatter radiation with wavelength in the range 1 to 4 µm. Hence, multiple scattering would increase the pathway of infrared light in the material, thus increasing the probability of radiations being absorbed. From the spectral data close to ignition, the apparent absorption coefficient has been calculated for the three studied samples. It leads respectively 127, 146 and 33 m$^{-1}$. These values remain lower than the 280 m$^{-1}$ required to correctly simulate time-to-ignition using the in-depth absorption model (see Section 3.1). To explain this small mismatch, it should be noted that thermo-radiative properties have been measured at room temperature, and the values may be higher in the pyrolysis range. Moreover, variations of thermophysical properties with temperature were not taken into account in the model, as conducted by other authors [10]. However, it can be concluded that bubbling has a significant influence on the thermo-radiative properties of PMMA that is likely to partially explain the delay of ignition observed when using halogen lamps as a heating source.

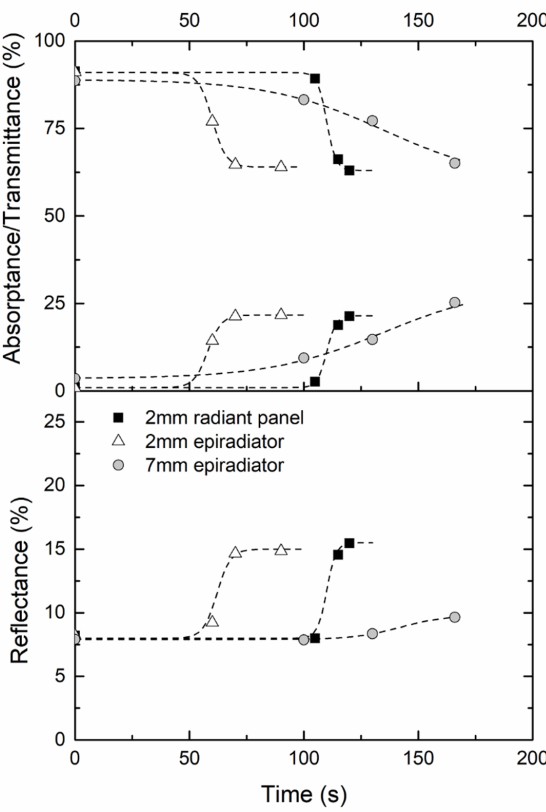

**Figure 14.** Evolutions of total transmittance, reflectance and absorptance evolutions according to the time of exposure for a PMMA sample (2 mm and 7 mm thickness) exposed to 27 kW/m$^2$ $\pm$ 2 kW/m$^2$.

## 4. Conclusions

Ignition of clear PMMA was studied using two different heating sources. It was shown that ignition time is closely related to the absorption spectrum of the polymer in the emission range of the source. With the near-infrared source, the ignition of clear PMMA was

strongly delayed, but it was nevertheless less than expected from its absorption coefficient in this domain. Since an intense bubbling of the sample was observed in the pre-ignition period, this phenomenon has been more accurately studied to understand its impact on the ignition time and the relation with the modification of the thermo-physical and/or thermo-radiative properties.

It was highlighted that the bubble size and number depend on the temperature gradient inside the sample itself dependent on: (i) exposure time, (ii) sample thickness, (iii) heat flux intensity and (iv) nature of the radiative heat source. Hence, the temperature gradient seems to govern the bubble size distribution: (i) a strong temperature gradient gives small bubbles formed in a thin layer underneath the surface, whereas (ii) a small temperature gradient contributes to the formation of small bubbles at the surface and larger bubbles in-depth.

Bubbling may affect ignition by modifying the thermo-physical and thermo-radiative properties of the polymer. The thermal diffusivity was evidenced to remain almost constant regardless of the bubbling degree in the polymer. This constant value may be explained by the decrease of thermal conductivity with bubbling that counterbalances the decrease of bulk density. Thus, the decrease of $k \cdot \rho \cdot C_p$ in bubbled samples is likely to induce a reduction of ignition time. Concerning the thermo-radiative properties, a drop of transmittance associated with an increase of absorptance in the near-infrared region was highlighted in samples containing bubbles. It may be assumed that the small bubbles located underneath the exposed surface and found in all samples could be responsible for the scattering of NIR radiations. Multiple scattering would increase the pathway of infrared light in the material, thus increasing the probability of radiations being absorbed. This phenomenon induces an increase in absorption coefficient that is also likely to reduce time-to-ignition when using halogen lamps (radiant panels) as a heating source.

We are conscious that our approach is not an in-situ study of bubbling. It gives a static view of bubbled sample morphology and properties are examined a posteriori at room temperature. Nevertheless, changes observed in both the thermo-physical and thermo-radiative properties of PMMA when bubbling provide reasonable explanation for ignition time modification. Future works will consist in considering the effect of temperature on the thermal properties.

**Supplementary Materials:** The following supporting information can be downloaded at: https://www.mdpi.com/article/10.3390/fire7040117/s1, Figure S1: Schematic representation of RAPACES (side view); Figure S2: Schematic representation of epiradiator with x is 30 mm (for a heat flux of 27 kW/m$^2$) or 40 mm (for a heat flux of 31 kW/m$^2$), and y is 10 mm.

**Author Contributions:** Conceptualization, G.R., L.A. and L.F., methodology, C.V., G.R., L.A., L.F. and C.L.; software, not applicable; validation, L.F. and L.A.; formal analysis, C.V. and L.F. investigation, C.V., L.A., P.S. and G.R.; resources, P.S.; data curation, C.V.; writing—original draft preparation, C.V. and L.F.; writing—review and editing, C.V., L.F., L.A., G.R. and C.L.; visualization, C.V. and C.L.; supervision, L.A., L.F. and C.L.; project administration, L.A.; funding acquisition, G.R. and L.A. All authors have read and agreed to the published version of the manuscript.

**Funding:** This research was funded by the French Alternative Energies and Atomic Energy Commission (CEA).

**Institutional Review Board Statement:** Not applicable.

**Informed Consent Statement:** Not applicable.

**Data Availability Statement:** The original contributions presented in the study are included in the article and Supplementary Material, further inquiries can be directed to the corresponding author.

**Acknowledgments:** The authors are grateful to the French Alternative Energies and Atomic Energy Commission (CEA) for the financial support and helpful collaboration on this research program. Authors are also thankful to the technicians of the IMT Mines Alès for technical aid during the development of RAPACES apparatus. Authors are grateful to Célia ANCER, Agnès BISTOLFI and Dominique LAFON of the IMT Mines Alès for experimental tests.

**Conflicts of Interest:** The authors declare no conflicts of interest.

## Nomenclature

| | |
|---|---|
| $a$ | Thermal diffusivity ($m^2 \cdot s^{-1}$) |
| $C_p$ | heat capacity ($J \cdot K^{-1} \cdot kg^{-1}$) |
| $e$ | thickness of sample (m) |
| $h_c$ | surface convective heat transfer coefficient ($W \cdot m^{-2} \cdot K^{-1}$) |
| $k$ | thermal conductivity ($W \cdot m^{-1} \cdot K^{-1}$) |
| $R$ | reflectance |
| $r$ | bubble radius (m) |
| $t$ | time (s) |
| $t_b$ | time of bubbling start (s) |
| $t_{ig}$ | time to ignition (s) |
| $T$ | temperature (K) |
| $T_0$ | initial temperature (K) |
| $T_b$ | bubbling temperature (K) |
| $T_{ig}$ | ignition temperature (K) |
| $T_S$ | surface temperature (K) |
| $x$ | in-depth distance (m) |
| $\alpha$ | absorptance (-) |
| $\Delta P$ | pressure variation (Pa) |
| $\varepsilon$ | surface emissivity |
| $\kappa$ | absorption coefficient ($m^{-1}$) |
| $\rho$ | material density ($kg \cdot m^{-3}$) |
| $\sigma$ | surface tension ($J \cdot m^{-2}$) |
| $\sigma_{SB}$ | Stefan Boltzmann constant ($W \cdot m^{-2} \cdot K^{-4}$) |
| $\theta$ | temperature rise (K), $\theta = T - T_0$ |
| $\tau$ | transmittance |

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
