# Peer review of "Effect of Bubbling on Ignition of PMMA Slab: Change in Thermo-Physical and Thermo-Radiative Properties"

_fire, doi:10.3390/fire7040117_

Round 1

Reviewer 1 Report

Comments and Suggestions for Authors

The team of authors conducted a thorough and carefully designed experiment to evaluate thermos-physical and thermo-radiative properties of PMMA, a standardized material which has been widely used to study characteristics of flammability, ignition and fire spreading. They studied the effect of external radiation sources on ignition, the formation of bubbles in PMMA samples, and how they affect material properties, such as thermal conductivity, density, and specific heat.

This reviewer offers two minor comments for the authors' consideration.  (1) How did the authors estimate the convective cooling or heating effect during the experiment, since it may influence ignition time?  (2) It would help readers to better understand their good experiment if they provide a schematic that can explain their experimental apparatus and methods.    

Overall, their experimental results are valuable for improving the currently existing thermal ignition model of PMMA and their experimental methods can also be applied to other flammable solids such as live and dead woods, which currently show some inconsistent data on ignition time.

Comments on the Quality of English Language

Their English is fine but minor editing can help as well. 

Author Response

The team of authors conducted a thorough and carefully designed experiment to evaluate thermos-physical and thermo-radiative properties of PMMA, a standardized material which has been widely used to study characteristics of flammability, ignition and fire spreading. They studied the effect of external radiation sources on ignition, the formation of bubbles in PMMA samples, and how they affect material properties, such as thermal conductivity, density, and specific heat.

This reviewer offers two minor comments for the authors' consideration. 

  1. How did the authors estimate the convective cooling or heating effect during the experiment, since it may influence ignition time? 

Response : thank you for this interesting question. Actually, cone calorimeter and radiant panel are tests with moderate ventilation (24L/s) and therefore the convection is low. In the literature, for these tests the coefficient of convection is often taken as 10 W/m²/K (Hopkins & Quintiere 1996). That is the value that we used in the modelling considering the Jiang model.

This clarification has been added in the revised manuscript.

  1. It would help readers to better understand their good experiment if they provide a schematic that can explain their experimental apparatus and methods.    

Response : Actually the schematic representation of experimental apparatus (RAPACES and epiradiator) was already provided in the supplementary material section. We don’t know if the reviewers have access to this section. You can see figures S1 and S2 at the end of this cover letter.

Overall, their experimental results are valuable for improving the currently existing thermal ignition model of PMMA and their experimental methods can also be applied to other flammable solids such as live and dead woods, which currently show some inconsistent data on ignition time.

Reviewer 2 Report

Comments and Suggestions for Authors

This work investigates the influence of bubbling on the thermo-physical and thermo-radiative properties of PMMA and the ignition of the polymer. The investigation is interesting but presents several questions that need to be clarified.

1.- The state-of-the-art needs to be improved because some studies were ignored.

2.- The authors must include an experimental setup section to describe the experiment in detail. There are some parts of the information in the results.

3.- The authors must explain the criterion to define the ignition point on the PMMA and how it was implemented in the formation of bubbles because the ignition changes.

4.- The results need to be improved because the discussion is focused on the comparison of measurements with other experiments. The discussion of results is poor. For instance, the authors propose that “bubbling may affect ignition by modifying the thermo-physical and thermo-radiative properties of the polymer.” This affirmation is correct because the polymer goes from being homogeneous to a heterogeneous material. The authors must propose more ideas.

5.- The conclusions are weak due to a lack of analysis of the results.

Comments on the Quality of English Language

Minor editing

Author Response

This work investigates the influence of bubbling on the thermo-physical and thermo-radiative properties of PMMA and the ignition of the polymer. The investigation is interesting but presents several questions that need to be clarified.

  • The state-of-the-art needs to be improved because some studies were ignored.

Response : Some additional articles have been included in the state of the art on PMMA ignition. However it should be noted that we did not find any new paper on PMMA bubbling which is the main subject of our article.

  • The authors must include an experimental setup section to describe the experiment in detail. There are some parts of the information in the results.

Response : Actually the schematic representation of experimental apparatus (RAPACES and epiradiator) was already provided in the supplementary material section.We don’t know if the reviewers have access to this section. You can see figures S1 and S2 at the end of this cover letter.

3.- The authors must explain the criterion to define the ignition point on the PMMA and how it was implemented in the formation of bubbles because the ignition changes.

Response : We are not sure to perfectly understand the question. Ignition is defined as the time the flame appears. Experimentally, it was visually detected with an accuracy of ±1s (This precision has been added in the text). As mentioned in the text, bubbling was studied in the pre-ignition period in order to understand its effect on the thermal properties and thus on ignition.

4.- The results need to be improved because the discussion is focused on the comparison of measurements with other experiments. The discussion of results is poor. For instance, the authors propose that “bubbling may affect ignition by modifying the thermo-physical and thermo-radiative properties of the polymer.” This affirmation is correct because the polymer goes from being homogeneous to a heterogeneous material. The authors must propose more ideas. The conclusions are weak due to a lack of analysis of the results.

Response : Considering our results on thermo-radiative and thermo-physical properties, the way bubbling affects ignition is detailed in page 17 and was a little emphasized in the revised manuscript to respond to reviewer : “It may be assumed that the small bubbles located underneath the exposed surface and found in all samples could be responsible for the scattering of NIR radiations as it has been shown in other weakly absorbing materials [44]. According to the Mie theory, bubbles in the radius range 50-200µm can scatter radiation with wavelength in the range 1 to 4µm. Hence, multiple scattering would increase the pathway of infrared light in the material thus increasing probability of radiations to be absorbed”.

We believe this explanation is consistent and relies on the results presented in the paper. Going further would be mere speculation, therefore we prefer to remain cautious.

Reviewer 3 Report

Comments and Suggestions for Authors

Dear authors....please find my review attached and I ask for only few clarifications

Author Response

Page 3 : what is Thermakin? A short hint or web link is useful.

Response : ThermaKin is a computational tool for modeling polymer pyrolysis and combustion behaviors developed by the Federal Aviation Administration (FAA) in the US. This precision has been added in the text.

Page 6 : how many replicates were done ?

Values indicated in the text are the average of at least 3 replicates. This has been added in the text.

Page 11: Please define what is critical radius rc ?

In the nucleation theory, the new phase that is formed (here the gas phase resulting from PMMA pyrolysis) appears as spherical nucleus. The critical radius rc corresponds to the radius above which the nucleus is stable and can grow to form a bubble. The definition has been added in the text.

Round 2

Reviewer 2 Report

Comments and Suggestions for Authors

The new version of the manuscript has been improved, but the authors must include an experimental setup section to describe the experiment in detail. There is information about the experiment in the results section. The discussion of results must be focused on the findings of the study, not on how the experiment was conducted.

Comments on the Quality of English Language

Minor editing

Author Response

The authors must include an experimental setup section to describe the experiment in detail. There is information about the experiment in the results section. The discussion of results must be focused on the findings of the study, not on how the experiment was conducted.

Response : Thank you for this remark. We have added some experimental details in the presentation of each setup. Moreover we have moved some experimental information from the section 3 (results) to the section 2 (Materials and methods). Hopefully, it is now convenient.